# Morphological Differences and Contour Visualization of Statoliths in Different Geographic Populations of Purpleback Flying Squid (*Sthenoteuthis oualaniensis*)

Moxian Chu [1], Bilin Liu [1,2,3,4,*], Liguo Ou [5], Ziyue Chen [1] and Qingying Li [1]

1   College of Marine Living Resource Sciences and Management, Shanghai Ocean University, Shanghai 201306, China; m220250829@st.shou.edu.cn (M.C.); zychen@shou.edu.cn (Z.C.); 51253904057@stu.ecnu.edu.cn (Q.L.)
2   National Engineering Research Center for Oceanic Fisheries, Shanghai 201306, China
3   The Key Laboratory of Sustainable Exploitation of Oceanic Fisheries Resources, Ministry of Education, Shanghai 201306, China
4   Key Laboratory of Oceanic Fisheries Exploration, Ministry of Agriculture and Rural Affairs, Shanghai 201306, China
5   School of Fishery, Zhejiang Ocean University, Zhoushan 316022, China; lgou@zjou.edu.cn
*   Correspondence: bl-liu@shou.edu.cn

**Abstract:** Statoliths are important hard tissues in cephalopods. Significant differences are found in the external morphology of statoliths in different groups or species. In this study, stepwise discriminant analysis was used to investigate the external morphological differences in purpleback flying squid statoliths in three different marine regions, comprising the East Indian Ocean (5° S–2° N, 82°–92° E), Central East Pacific Ocean (02°37′ S–0°59′ N, 99°44′ W–114°19′ W), and Northwest Indian Ocean (17°04′ N–17°18′ N, 61°05′ E–61°32′ E). The contours of statoliths were reconstructed visually by using Fourier analysis and the landmark method. The results obtained by stepwise discriminant analysis showed that the accuracy of identification was 84.4% for the traditional measurement method, 82.9% for the Fourier analysis method, and 87.3% for the landmark method. The contour visualization results showed that the purpleback flying squid statoliths were small in the Central East Pacific Ocean, and the curvature of the side region was the most obvious. The radian differentiation of statoliths was most gentle in the East Indian Ocean. In the Northwest Indian Ocean, the rostral region of statoliths was shorter and the dorsal region was smoother. The reconstruction results detected significant differences in the outer morphology of statoliths in different marine regions. The results obtained in this study show that all three methods are effective for identifying populations, but the landmark method is better than the traditional measurement method. The reconstruction of statolith contours using the Fourier transform and landmark methods provides an important scientific basis for conducting taxonomy, according to statolith morphology.

**Keywords:** contour visualization; elliptic Fourier transform; landmark method; population identification; purpleback flying squid; statolith morphology

## 1. Introduction

The purpleback flying squid (*Sthenoteuthis oualaniensis*) belongs to the class Cephalopoda, family *Ommastrephidae*, and genus *Sthenoteuthis*. It is widely distributed in the tropical and subtropical waters of the Indian Ocean and Pacific Ocean [1]. In resource surveys conducted from the 1960s to the 1980s, the purpleback flying squid was identified as having high abundance in the northwest Indian Ocean and Arabian Sea [2]. In the Central East Pacific, its geographic distribution often overlaps with that of the jumbo (flying) squid (*Dosidicus gigas*), and they are typically caught together in fishery operations [3]. Initially, China conducted resource surveys on the purpleback flying squid resources in the northwest Indian Ocean from 2003 to 2005 [4]. It was estimated that the total biological

biomass of squid in the Indian Ocean ranged from 3 to 4.2 million tons [1]. Despite the abundant squid resources, there is a high potential for exploitation. As a result, there is a possibility that the future development level may be sustained at a relatively high level [1,4].

The hard tissues of cephalopods, including statoliths, beaks, internal shells, etc., have stable structures, and their continuous growth throughout their life is considered a reliable information carrier [5]. The statolith, as one of the hard tissues in cephalopods, is an essential component of the cephalopod acceleration sensing system [6]. Due to its calcified structure, the morphology of the statolith is not easily damaged [7], and it is highly specific [8]. Therefore, they are frequently employed in conjunction with beaks, internal shells, eye lenses, and other hard tissues for studying age and growth [9–11], species identification [12], population differentiation [13–15], and other purposes [7,8,16].

Statolith morphology is quantitatively analyzed based on the external shape. Thus, the ecological characteristics of fisheries can be investigated based on variations in the external shape of statoliths and relevant biological information. The two main methods used for analyzing statolith morphology are traditional morphometry and geometric morphometry. Traditional morphometry is the most widely used method, and it involves measuring statolith parameters, such as the total length. This method has been applied to analyze the external morphology of statoliths in various areas for *Dosidicus gigas* [17] and to study growth patterns based on morphological differences in the statoliths of this squid [18]. Initially, studies on statolith morphology were mostly based on linear distance measurement [19,20]. However, traditional measurement methods can introduce experimental errors [21] and fail to accurately reconstruct the original shape [22].

Geometric morphometrics can effectively address this issue [23]. Significant progress has been made in the morphological study of statoliths [24–29]. Efficient and simple morphological methods can be applied to analyze fish otoliths for individual fish identification, detecting interspecific differences, and population discrimination [14,30,31]. In previous studies, geometric morphometry analysis has also been employed to investigate fish otoliths and other hard tissues in fish. In addition, the combination of Fourier analysis with traditional morphometric methods has proven effective for classifying species based on the morphological features of fish otoliths [32–34]. Moreover, landmark methods are highly useful for assessing statolith morphology. In comparison to Fourier analysis, they can directly capture and describe the shape features of objects. The handling of details and local features can be demonstrated by selecting different landmarks [33,35,36]. Statoliths are sometimes combined with other structures (such as the body or the beak) for discriminant analysis [14,37].

Previous studies of statoliths in the purpleback flying squid mainly focused on their microstructure [38–40], microchemistry [41], growth characteristics [18], morphological analysis [42–44], and the effects of life and growth patterns [24]. Studies regarding the discrimination of different geographical populations of purpleback flying squid based on statolith morphology mostly used traditional morphometry [42], whereas few employed Fourier transform and landmark methods [32]. Therefore, in the present study, statolith samples from purpleback flying squid collected in the Northwest Indian Ocean, East Indian Ocean, and Central East Pacific Ocean were analyzed using three methods, comprising traditional measurement, Fourier transform, and landmark methods. These methods were used to analyze differences in the morphological structural characteristics of statoliths among three different populations and to assess the feasibility of population discrimination. The results obtained in this study can help distinguish different purpleback flying squid populations using statoliths. In particular, reconstructing statolith profiles from different regions using Fourier transform and landmark methods can address the shortcomings of incomplete data in traditional morphometrics.

## 2. Materials and Methods

### 2.1. Materials

Experimental purpleback flying squid samples were collected from the East Indian Ocean, Northwest Indian Ocean, and Central East Pacific Ocean in November 2019, June 2019, and February 2018, respectively, and the collection areas were at the following coordinates: 5° S–2° N, 82°–92° E; 17°04′–17°18′ N, 61°05′–61°32′ E; and 02°37′ S–0°59′ N, 99°44′–114°19′ W (Table 1). The sampling distribution map is shown in Figure 1. The research indicates that there is no statistically significant difference in morphometrics between the left and right statoliths [15]. Therefore, after conducting basic biological measurements for the purpleback flying squid, the left statolith was extracted as the experimental sample. In total, 55 statolith samples were obtained from the East Indian Ocean, 63 from the Central East Pacific Ocean, and 55 from the Northwest Indian Ocean.

**Table 1.** Sampling areas and sample details.

| Sea Area | Sampling Area | Sampling Date | Number of Samples | Mantle Length/mm |
|---|---|---|---|---|
| Central East Pacific Ocean | 02°37′ S–0°59′ N 99°44′–114°19′ W | Feburary 2018 | 63 | 121–427 |
| Northwest Indian Ocean | 61°05′–61°32′ E 17°04′–17°18′ N | Feburary 2019 | 55 | 142–525 |
| East Indian Ocean | 82°–92° E 5°S–2° N | November 2019 | 55 | 80–176 |

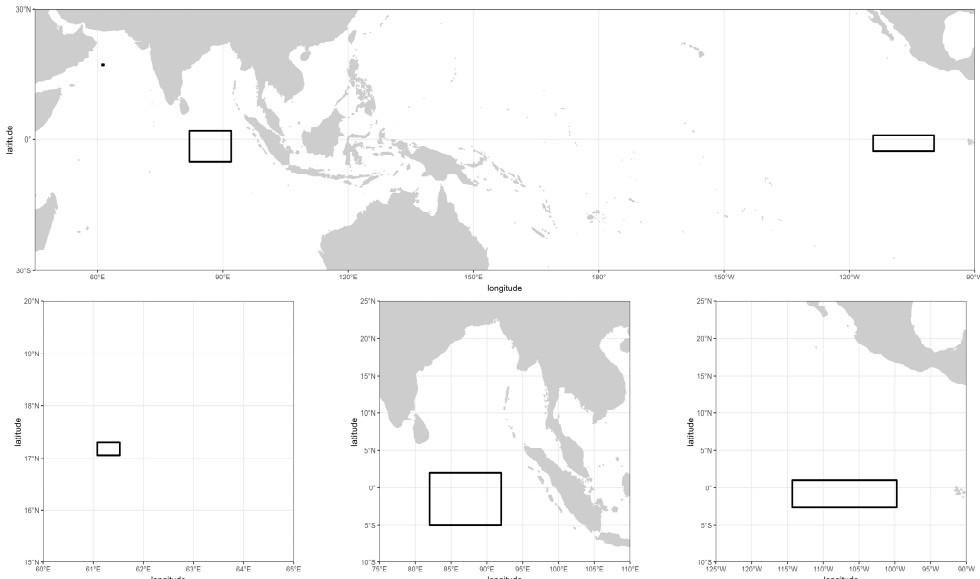

**Figure 1.** Distribution of sampling regions.

### 2.2. Methods

2.2.1. Sample Handling

The statolith samples collected from purpleback flying squid in three marine areas were cleaned with alcohol and then with an ultrasonic cleaner. Next, they were thoroughly rinsed with distilled water and placed in a drying oven (temperature: 50–60 °C) until no changed occurred in the weight of the statoliths.

2.2.2. Image Acquisition and Morphological Determination of Statoliths

The prepared statoliths were placed under a microscope for observation. Images were captured using an optical microscope (Olympus Model BX51, Olympus, Tokyo, Japan) at 100× magnification. The captured photos were processed using Photoshop CS 6.0 (Adobe

Systems Inc., San Jose, CA, USA, 2019) to adjust the brightness and contrast and to remove any artifacts [33]. The morphological parameters measured for the statoliths comprised the total statolith length (TSL), maximum width (MW), lateral dome length (LDL), dorsal dome length (DLL), rostrum lateral length (RLL), rostrum length (RL), rostrum width (RW), statolith wing length (WL), and statolith wing width (WW) [44]. The data (Figure 2) obtained were used for discriminant analysis.

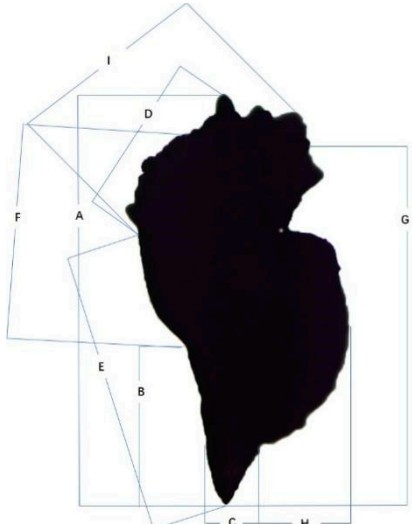

**Figure 2.** Scheme of morphological parameters of statolith. NOTE: A—TSL; B—RL; C—RW; D—DLL; E—RLL; F—LDL; G—WL; H—WW; I—MW.

### 2.2.3. Elliptical Fourier Analysis

SHAPE software was employed according to the methods described in a previous study [28]. Two-dimensional images of statoliths were processed by grayscale conversion, image binarization, noise elimination, chain code extraction, and Fourier transformation to obtain multiple Fourier feature coefficients representing the statolith shapes. The differences between statolith shapes were distinguished by analyzing these feature coefficients.

Previous studies [26,29] suggest that 20 harmonics can adequately describe the morphological contours of statoliths. Hence, the first 20 harmonics were extracted for discriminant analysis in the present study. The program extracted the data variables for subsequent discriminant analysis.

### 2.2.4. Landmark Analysis

In biology, morphological description often involves the use of landmark methods, where two-dimensional images are converted into data points on a coordinate axis by placing landmarks on them. There are three types of landmarks in total. According to previous studies, 14 landmarks were selected based on the morphological features of statoliths (Figure 3). Among these landmarks, type I landmarks comprising 2, 8, 10, 11, and 13 represented critical points in various regions of the statolith, type II landmarks consisting of 3, 4, 5, 6, and 7 denoted depressions and protrusions in the wing area of the statolith, and type III landmarks comprising 1, 9, 12, and 14 corresponded to the highest, widest, and outermost points of the statolith, respectively.

After marking these landmarks using tpsDig2 ver.2.31 software (Figure 3), least-squares regression analysis was conducted using tpsSmall ver1.34 software. The results showed that the regression coefficient between the Procrustes distance (x-axis) and the distance in tangent space (y-axis) was 0.9993, which is close to 1, thereby indicating that the landmarks selected for this study were effective.

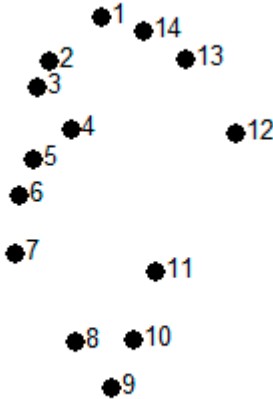

**Figure 3.** Mean shape. The numbers represent the landmarks' identifiers.

### 2.3. Data Analysis

Statistical analysis was conducted based on the nine morphological parameters, 77 Fourier coefficients, and relative warp scores obtained from 14 landmarks for the statoliths from purpleback flying squid collected in three marine regions. Independent pairwise sample *t*-tests were performed for each of the three marine regions based on the nine morphological parameters, to detect significant differences. Stepwise discriminant analysis was applied separately to the nine morphological parameters, 77 standardized Fourier coefficients, and relative warp scores. A stepwise method and Wilks' lambda test were used to establish discriminant functions by eliminating coefficients that did not contribute significantly to discrimination. Scatter plots were generated for discriminant analysis. All data analysis and graphing processes were conducted using SPSS 25.0, R 4.3.1, and Excel 2016 software.

## 3. Results

### 3.1. Morphological Differences

Differences were analyzed using *t*-tests after comparing the statolith morphological parameters for the three marine regions (Table 2), and the results are shown in Table 3. In the East Indian Ocean and Central East Pacific Ocean regions, significant differences were found in eight morphological parameter indicators ($p < 0.01$), excluding WW. Only two indicators ($p < 0.05$), comprising WW and MW, differed between the East Indian Ocean and Northwest Indian Ocean regions. In contrast, all of the indicator parameters differed significantly between the Northwest Indian Ocean and Central East Pacific Ocean regions ($p < 0.01$).

**Table 2.** Comparison of morphological parameters for purpleback flying squid statoliths in three marine regions.

| Area | East Indian Ocean | Northwest Indian Ocean | Central East Pacific Ocean |
|---|---|---|---|
| **Parameters** | **Min-Max Value (Mean ± S.D)** | **Min-Max Value (Mean ± S.D)** | **Min-Max Value (Mean ± S.D)** |
| TSL | 1652.55–1034.90 (1241.06 ± 143.11) | 1464.39–906.87 (1262.11 ± 142.52) | 1144.41–762.45 (1014.31 ± 73.98) |
| RL | 582.09–303.28 (427.22 ± 61.80) | 503.55–267.74 (402.78 ± 52.28) | 447.51–285.26 (366.05 ± 42.99) |
| RW | 262.54–131.87 (185.11 ± 27.88) | 265.26–118.11 (182.14 ± 37.71) | 183.76–104.79 (135.54 ± 15.93) |
| DLL | 964.60–456.03 (591.65 ± 95.42) | 782.89–405.45 (625.27 ± 104.93) | 518.90–302.77 (439.23 ± 49.25) |

**Table 2.** *Cont.*

| Area | East Indian Ocean | Northwest Indian Ocean | Central East Pacific Ocean |
|---|---|---|---|
| **Parameters** | **Min-Max Value (Mean $\pm$ S.D)** | **Min-Max Value (Mean $\pm$ S.D)** | **Min-Max Value (Mean $\pm$ S.D)** |
| RLL | 1151.39–721.36 (868.97 $\pm$ 110.47) | 1041.52–618.51 (863.80 $\pm$ 101.90) | 881.14–570.47 (722.54 $\pm$ 64.37) |
| LDL | 1020.69–575.71 (739.38 $\pm$ 90.09) | 947.61–514.31 (782.51 $\pm$ 107.55) | 715.98–451.64 (599.82 $\pm$ 54.32) |
| WL | 1431.86–884.87 (1073.34 $\pm$ 127.25) | 1262.26–768.16 (1047.48 $\pm$ 115.48) | 1010.95–663.46 (862.50 $\pm$ 70.01) |
| WW | 312.65–131.68 (203.36 $\pm$ 43.43) | 468.33–183.84 (298.03 $\pm$ 62.69) | 327.69–112.35 (217.59 $\pm$ 47.96) |
| MW | 1015.16–614.59 (730.18 $\pm$ 76.33) | 931.80–584.02 (782.25 $\pm$ 95.61) | 694.77–417.50 (593.23 $\pm$ 60.29) |

**Table 3.** Results of *t*-tests based on morphological parameters.

| Parameters | East Indian Ocean & Central East Pacific Ocean | | East Indian Ocean & Northwest Indian Ocean | | Northwest Indian Ocean & Central East Pacific Ocean | |
|---|---|---|---|---|---|---|
| | **t** | *p* | **t** | *p* | **t** | *p* |
| TSL | 9.427 | <0.01 | −0.625 | >0.05 | −10.075 | <0.01 |
| RL | 5.287 | <0.01 | 1.83 | >0.05 | −3.678 | <0.01 |
| RW | 10.247 | <0.01 | 0.372 | <0.05 | −7.39 | <0.01 |
| DLL | 9.509 | <0.01 | −1.413 | >0.05 | −10.448 | <0.01 |
| RLL | 6.796 | <0.01 | 0.207 | >0.05 | −7.705 | <0.01 |
| LDL | 8.76 | <0.01 | −1.823 | <0.05 | −9.891 | <0.01 |
| WL | 8.578 | <0.01 | 0.908 | <0.05 | −8.99 | <0.01 |
| WW | −1.359 | <0.05 | −7.282 | <0.05 | −6.933 | <0.01 |
| MW | 9.034 | <0.01 | −2.517 | <0.05 | −10.993 | <0.01 |

Note: $p < 0.01$ denotes a highly significant difference; $p < 0.05$ denotes a significant difference; $p > 0.05$ denotes no significant difference.

### 3.2. Discriminant Analysis Using Different Methods

Stepwise discriminant analysis was performed using the statolith's morphological measurements to classify the purpleback flying squid in three marine regions. Among the nine statolith morphological parameters, four indicators (LDL, WL, WW, and RW), all with *p*-values < 0.05, were selected for the final discriminant analysis. Cross-validation is performed exclusively on cases under analysis. In cross-validation, each case is classified by functions derived from all cases except for that particular case. Cross-validation of the discriminant analysis results showed that the purpleback flying squid statolith morphology classification accuracies for the East Indian Ocean, Central East Pacific Ocean, and Northwest Indian Ocean regions were 69.1%, 95.2%, and 87.3%, respectively, with an overall accuracy of 84.4% (Table 4). Discriminant function plots effectively differentiated between purpleback flying squid populations, with minimal overlap observed in discriminant function 2, which distinguished between the East Indian Ocean, Central East Pacific Ocean, and Northwest Indian Ocean regions. Discriminant function 1 significantly separated Central East Pacific Ocean purpleback flying squid from Northwest Indian Ocean purpleback flying squid, and the only misclassification occurred between the East Indian Ocean and Northwest Indian Ocean regions (Figure 4).

**Table 4.** Cross-validation results for discriminating populations of purpleback flying squid in three marine regions.

| Method | Pre-Discrimination Species | Discriminated Species | | | | Discrimination Accuracy (%) |
|---|---|---|---|---|---|---|
| | | East Indian Ocean | Central East Pacific Ocean | Northwest Indian Ocean | Sample Size | |
| Traditional morphometric method | East Indian Ocean | 38 (69.1%) | 7 (12.7%) | 10 (18.2%) | 55 (100%) | 84.40% |
| | Central East Pacific Ocean | 3 (4.8%) | 60 (95.2%) | 0 (0%) | 63 (100%) | |
| | Northwest Indian Ocean | 6 (10.9%) | 1 (1.8%) | 48 (87.3%) | 55 (100%) | |
| Fourier analysis method | East Indian Ocean | 38 (69.1%) | 7 (12.7%) | 10 (18.2%) | 55 (100%) | 82.10% |
| | Central East Pacific Ocean | 5 (7.9%) | 54 (85.7%) | 4 (6.3%) | 63 (100%) | |
| | Northwest Indian Ocean | 2 (3.6%) | 3 (5.5%) | 50 (90.9%) | 55 (100%) | |
| Landmark-based method | East Indian Ocean | 44 (80.0%) | 9 (16.4%) | 2 (3.6%) | 55 (100%) | 87.30% |
| | Central East Pacific Ocean | 5 (7.9%) | 57 (90.5%) | 1 (1.6%) | 63 (100%) | |
| | Northwest Indian Ocean | 1 (1.8%) | 4 (7.3%) | 50 (90.9%) | 55 (100%) | |

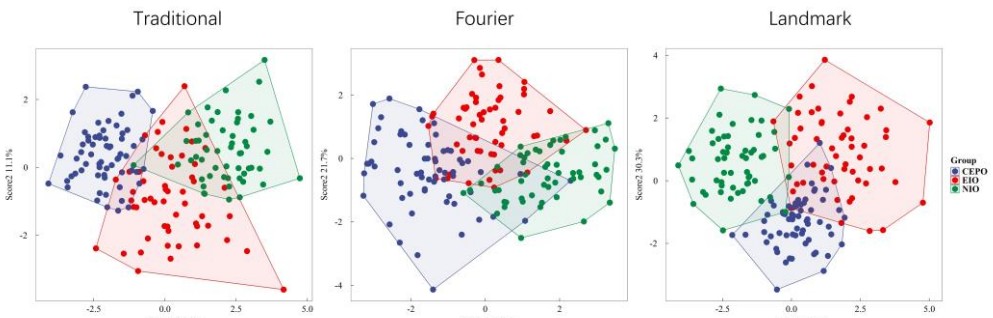

**Figure 4.** Scatterplot of typical discriminant functions for overall morphology of purpleback flying squid statoliths in three marine regions. The three methods, from left to right, are the traditional measurement method, Fourier analysis method, and landmark method. (CEPO = Central East Pacific Ocean, EIO = East Indian Ocean, NIO = Northwest Indian Ocean).

The results obtained by stepwise discriminant analysis based on Fourier analysis showed that among the 77 Fourier coefficient indicators representing the shape profiles of the statoliths, only nine indicators (d1, d5, a19, a5, d3, d17, d2, a9, and b4) were used in the final discriminant analysis. Cross-validation showed that the purpleback flying squid statolith morphology classification accuracies for the East Indian Ocean, Central East Pacific Ocean, and Northwest Indian Ocean regions were 84.4%, 78.0%, and 87.8%, respectively, with an overall accuracy of 82.90% (Table 4). The discriminant function plots had similar patterns to those obtained using traditional morphometric measurements, with an overall effective discrimination effect (Figure 4).

Using the 24 relative warp scores obtained with tpsRelw ver.1.75 software, nine variables ($p < 0.05$) were included in the discriminant analysis: RW1, RW4, RW9, RW2, RW5, RW13, RW3, RW12, and RW10. Cross-validation showed that the purpleback flying squid statolith morphology classification accuracies for the East Indian Ocean, Central East Pacific Ocean, and Northwest Indian Ocean regions were 90.9%, 90.5%, and 80.0%, respectively, with an overall accuracy of 87.30% (Table 4). The discriminant function plots showed that discriminant function 1 effectively separated Northwest Indian Ocean purpleback flying squid from East Indian Ocean flying squid. Discriminant function 2 distinguished Central East Pacific flying squid from those in the other two regions, thereby resulting in clear separation of the three marine regions. Overall, the discrimination effect was quite satisfactory (Figure 4).

### 3.3. Reconstruction of Purpleback Flying Squid Statolith Morphology

3.3.1. Statolith Morphology Reconstruction Based on Fourier Transform

The application of Fourier transform to reconstruct the statolith morphology for purpleback flying squid from three marine regions characterized the overall shape of the statoliths as elongated. The dorsal region exhibited irregular protrusions and depressions, whereas the lateral region was inclined and relatively long, with a smoother edge. The rostral region was narrow and elongated, whereas the wing area was broad, which aligns with the habitat preferences of purpleback flying squid in the upper–middle water layers. A noticeable groove separated the dorsal region from the wing region.

For the three populations of purpleback flying squid, Fourier harmonics ranging from 1 to 5 formed the initial structure of the dorsal, lateral, rostral, and wing regions. From 6 to 10, significant changes occurred in the wing and rostral regions, with noticeable grooves apparent in the dorsal region. From 11 to 20, the changes in statoliths manifested as subtle variations in localized regions, and the statolith shape closely resembled the actual contour of the statolith at Fourier harmonic 20 (Figure 5).

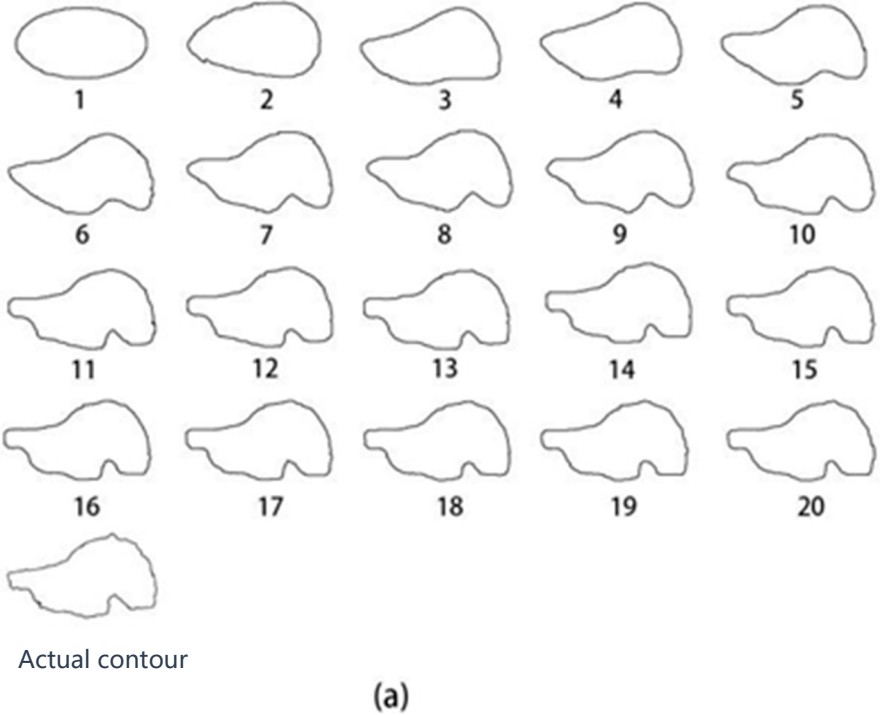

**Figure 5.** *Cont.*

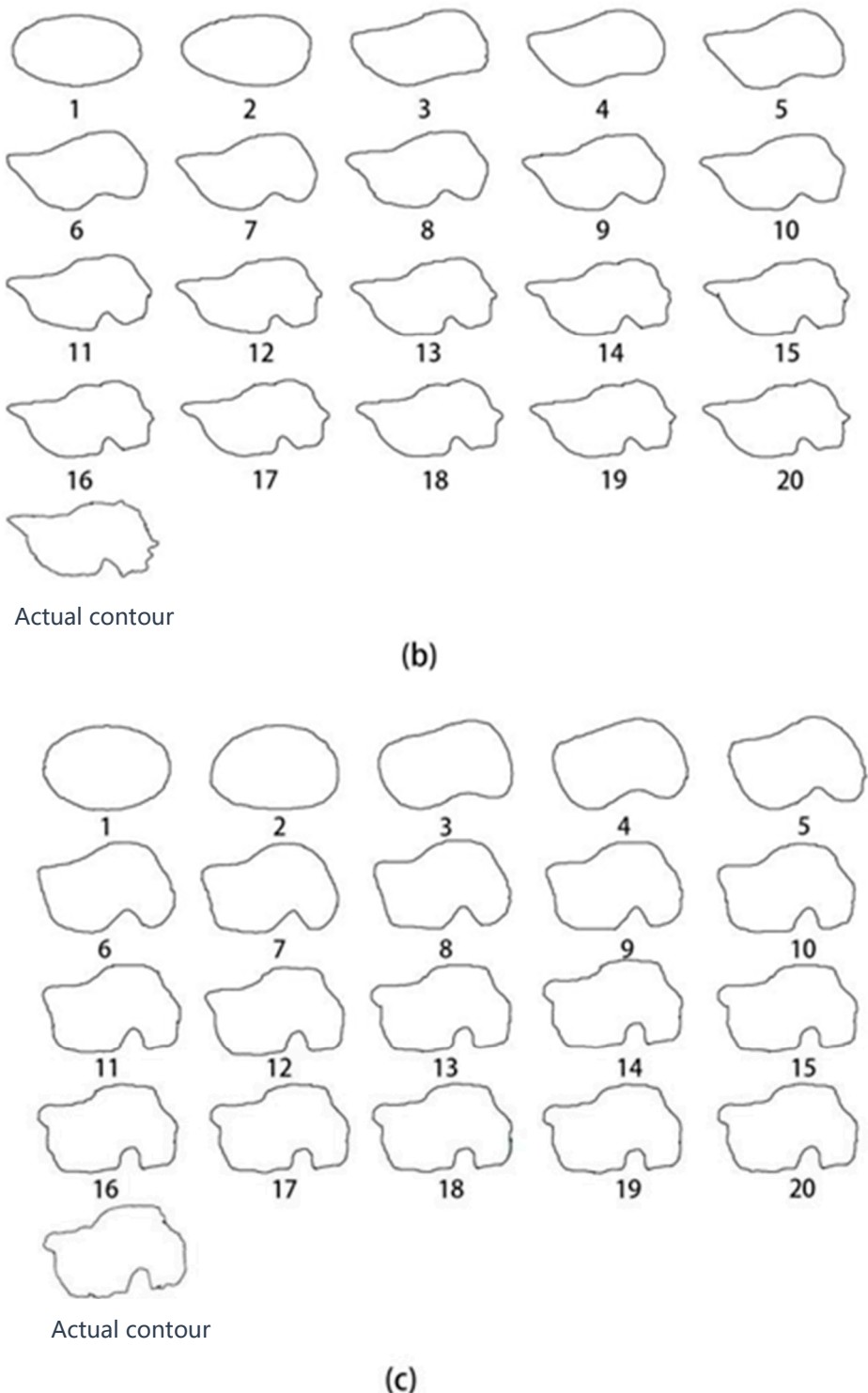

Actual contour

**(b)**

Actual contour

**(c)**

**Figure 5.** Reconstruction of entire statolith shape and sulcus shape from the number of Fourier harmonics. Note: The number (1–20) represents the number of harmonics used for the shape. (**a**) Statolith morphology in East Indian Ocean. (**b**) Statolith morphology in Central East Pacific Ocean. (**c**) Statolith morphology in Northwest Indian Ocean.

### 3.3.2. Statolith Morphology Reconstruction Using the Landmark Method

The analysis described above demonstrated that both type I and type II landmarks contributed significantly, where they primarily manifested in the dorsal and wing areas of the statolith. Mesh deformation maps and vector deformation maps were generated using

tpsRegr ver.1.45 software for average form visualization (Figures 6 and 7). The statoliths from the Northwest Indian Ocean clearly exhibited more pronounced changes compared with those from the other two regions. In the statoliths from the Northwest Indian Ocean, type II landmarks 5 and 6 were clearly sunken, relative to those in the statoliths from the East Indian Ocean and Pacific regions, whereas type I landmark 8 was elevated, and type III landmark 9 was contracted. Compared with statoliths from the East Indian Ocean and Northwest Indian Ocean regions, the statoliths from the Pacific region were characterized by sunken type I landmarks 11 and 13 and elevation of type III landmark 14. The visualized forms of the landmarks resembled the statolith shapes, and the morphological differences between populations were clearly discernible.

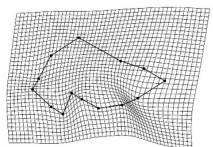 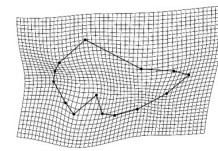 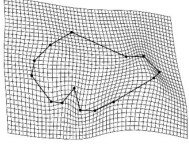

**Figure 6.** Variations enlarged three times in the vector diagram for landmarks. Note: In the following order: East Indian Ocean, Central East Pacific Ocean, and Northwest Indian Ocean.

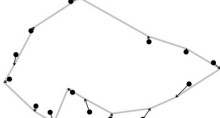 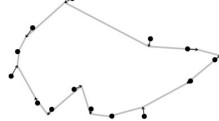 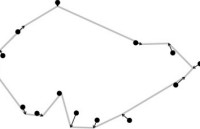

**Figure 7.** Variations enlarged three times in the diagram for landmarks. Note: In the following order: East Indian Ocean, Central East Pacific Ocean, and Northwest Indian Ocean.

## 4. Discussion

### 4.1. Comparative Analysis of Three Methods for Population Discrimination

Statolith morphology is very useful for discriminating populations, and it provides the basis for studying the fisheries ecology of different groups [7,45,46]. Identification of different populations within the same species is challenging [47]. In the present study, statoliths from three different marine regions were subjected to discriminant analysis using three morphological methods. Classification accuracy is improved by utilizing stepwise discriminant analysis with variables that exhibit significant differences [14,45]. The results indicate that all three methods can successfully discriminate the morphologies of statoliths of the purpleback flying squid from different marine areas.

The accuracy of purpleback flying squid statolith morphology classification was 95.2% in the Pacific region, thereby indicating relatively high performance. However, the accuracy of purpleback flying squid statolith morphology classification was less than 80% in the East Indian Ocean region, with much less accurate discrimination, as shown clearly by the discriminant function scatter plots.

In Fourier analysis, nine Fourier harmonics were used for the final population discriminant analysis, with a success rate of 82.9%, according to cross-validation. According to the discriminant function scatter plots, the external morphology could be effectively distinguished for purpleback flying squid statoliths from the three marine regions. Previous studies of differences in the external morphology of statoliths from fish species have indicated that both traditional morphological and Fourier analysis methods can be used for identification, but Fourier analysis is more reliable [14,15]. In the present study, there was little difference between the two methods, thereby demonstrating that both can be used for discriminating purpleback flying squid statoliths.

In a previous study, the landmark method demonstrated good performance in discriminating fish body morphology among different Yellow River carp lineages [48]. There are also relevant applications in cephalopods [12]. In the present study, nine variables

were used in the final discriminant analysis for the landmark method, and the classification accuracy exceeded 87% for all three populations, with high overall accuracy. The discrimination results were superior to those using the other two methods, and population differences were clearly detected. Previous studies also showed that traditional morphometric measurement, Fourier analysis, and landmark methods were effective in classifying and identifying the morphology of fish otoliths [33], and similar results were obtained in the present study.

### 4.2. Analysis of Morphological Differences in Statoliths among Different Populations

Reconstructing the morphology of cephalopod statoliths can visualize the differences in statolith morphology among different populations. The variations in statolith morphology among different populations may be associated with geographic origin [49].

In the present study, Fourier analysis and the landmark method were employed for the morphological reconstruction of statoliths from different marine regions. Although obtaining harmonics is highly complex, Fourier analysis is a relatively reliable analytical method because it eliminates the effects of factors such as position, size, and orientation when describing changes in the morphological contours of statoliths and individual differences [50]. The final discriminant analysis used nine of the 77 Fourier harmonics from the statolith profiles in the three populations. Fourier analysis showed that the first five harmonics indicated larger wing regions in the statoliths from the Northwest Indian Ocean region compared with the other two regions. Thus, the first five harmonics contributed significantly to purpleback flying squid population discrimination, and similar results were obtained in previous studies of fish identification [26]. In particular, thin-plate spline analysis was used by Bookstein [51] to visualize the deformation degree of points in the mesh deformation plot, and points with greater distortion were considered points with higher variability. For statoliths from the Northwest Indian Ocean region, type II landmarks 5 and 6 were characterized as sunken compared with those from the East Indian Ocean and Central East Pacific regions, whereas type I landmark 8 was elevated, and type III landmark 9 was contracted. These results indicate that the anterior ends of statoliths from the Northwest Indian Ocean region were shorter in the rostrum region, with a larger wing region, and consistent results were obtained by Fourier analysis. For statoliths from the Central East Pacific region, statolith points 12 and 13 were sunken, whereas point 14 was elevated (Figure 7), thereby indicating noticeable curvature in the lateral region of the statolith. Visualizing the statolith contours using both methods obtained more complete restoration of the original statolith morphology. These results show that the statolith structure in purpleback flying squid is similar to that in other members of *Ommastrephidae*, with distinct structures in the dorsal, wing, lateral, and rostrum regions [44]. The large wing region and narrow rostrum region indicate that purpleback flying squid is a species that inhabits pelagic water [24].

Purpleback flying squid from different marine regions may exhibit variations due to their different habitats [52]. Different movement patterns can result in varying morphologies of statoliths in their statocyst [53]. In the reconstructed images of statoliths in this study (Figure 7), the most significant differences were observed in the wing and rostrum regions. These two parts play a crucial role in regulating acceleration during the movement of pelagic squids [54]. The population from the Northwest Indian Ocean, as described in this study, exhibits higher sensitivity to low acceleration compared to the other two regions. Large statoliths enhance the flow of lymph, which makes them more sensitive to lower acceleration, thereby resulting in higher overall sensitivity by the acceleration sensory system, whereas the opposite applies to small statolith [6,24].

In the present study, purpleback flying squid statoliths from the Central East Pacific were noticeably smaller than those from the two Indian Ocean regions, possibly due to differences in the ocean currents between the Pacific and Indian Oceans [43,54,55]. Different ocean currents lead to distinct movement patterns and feeding behaviors in purpleback flying squid [53]. The Northwest Indian Ocean is mainly influenced by upwelling caused

by surface winds, which brings oxygen-deficient but nutrient-rich deep waters to the surface [56]. During the sampling season in November, the East Indian Ocean experiences a transition between monsoon periods, and an equatorial current runs from west to east near the equator. Meanwhile, south of the equator, there is an east-to-west ocean current known as the South Equatorial Current, along with boundary currents to the east and west. Together, they form a clockwise closed-loop circulation [54,56]. The Central East Pacific is primarily influenced by the warmer Equatorial Countercurrent and cooler South Equatorial Current, leading to lower primary productivity [43]. The different flow patterns in various marine regions influence their swimming behavior, which, in turn, affects the morphology of their statoliths.

## 5. Conclusions

In the present study, morphometric, Fourier analysis and landmark methods were employed to conduct discriminant analysis for statoliths from purpleback flying squid from three different marine regions and to reconstruct statolith morphology. The results demonstrated the effectiveness of all three methods for population discrimination. Compared with traditional morphometric methods, Fourier analysis and landmark methods were more effective in discriminating between populations from different marine regions. Traditional morphometric methods obtained less satisfactory performance in discriminating between samples from the Indian Ocean and Northwest Indian Ocean, possibly due to the close geographical proximity of the sampling locations and minor differences in statolith morphology. The landmark method is currently one of the most widely used techniques in geometric morphometrics. The success rate of the landmark method in discrimination was higher than those of the other two methods in the present study, which indicates that appropriate landmarks were selected, and thus, the completeness of statolith morphology was ensured, while also avoiding errors introduced by irrelevant information. Therefore, the landmark method is advantageous for interpreting morphological variations and providing biological explanations for differences.

In this study, elliptical Fourier analysis was also employed to reconstruct the statolith contours, and the results confirmed the effectiveness of using elliptical Fourier descriptors for accurately characterizing statolith contour shapes. The landmark method was used to visualize the variations in statolith feature points, and the changes were illustrated as vector plots. These findings provide a significant scientific foundation for the morphological classification of statoliths.

Currently, the analysis of statoliths in fisheries relies primarily on two-dimensional images. Compared to three-dimensional images, two-dimensional images may miss morphological differences [57]. In addition, artificial intelligence is the current development trend [58]. In the future, three-dimensional image recognition and automated landmark point extraction can be applied to classification. Combining artificial intelligence with morphological methods will reduce manual labor costs and experimental errors and potentially lead to significant advances in this field.

**Author Contributions:** B.L.: Conceptualization; resources; writing—review and editing. M.C.: Conceptualization; methodology; formal analysis; investigation; writing—original draft; software. L.O.: Methodology; software; writing—review. Z.C.: Writing—review and editing. Q.L.: Methodology. All authors have read and agreed to the published version of the manuscript.

**Funding:** This work was sponsored by the Nation Key R&D Program of China (2023YFD2401302); Follow-up program for the Professor of Special Appointment (Eastern Scholar) at Shanghai Institutions of Higher Learning under Contract (GZ2022011); Monitoring and Assessment of Global Fishery Resources (comprehensive scientific survey of fisheries' resources at the high seas).

**Institutional Review Board Statement:** Not applicable.

**Informed Consent Statement:** Not applicable.

**Data Availability Statement:** Data are contained within the article.

**Conflicts of Interest:** The authors declare no conflicts of interest.

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
