# Peer review of "Morphological Differences and Contour Visualization of Statoliths in Different Geographic Populations of Purpleback Flying Squid (Sthenoteuthis oualaniensis)"

_jmse, doi:10.3390/jmse12040597_

Round 1

Reviewer 1 Report

Comments and Suggestions for Authors

Comments and suggestions for authors

Main comments:

I think this work is very interesting and should be published

I suggest to the Authors to improve the exposure of the text with minor revisions.

Suggestions

Introduction

line 50: Insert “l” in the word level

line 51-53: The authors should enrich the sentences on the enormous importance of hard structures for the study of Cephalopod.

Please also mention at the end the following papers about hard structures:

Ø Arkhipkin, A.I., Bizikov, V.A., Doubleday, Z.A., Laptikhovsky, V.V., Lishchenko, F.V., Perales-Raya, C., et al. (2018) Techniques for estimating the age and growth of Molluscs: Cephalopoda. J. Shellfish Res. 37(4), 783–793

Ø Agus, B., Mereu, M., Cannas, R., Cau, A., Coluccia, E., Follesa, M.C., et al. (2018) Age determination of Loligo vulgaris and Loligo forbesii using eye lens analysis. Zoomorphology 137, 63-70. DOI: 10.1007/s00435-017-0381-8

Materials and Methods.

line 108 : Only the left (why only one?) Is there a specific reason?

Results:

lines 177-19: Paragraph “ 3.1. Morphological differences”

I strongly suggest that the authors include a photograph of the statoliths (either in pairs left and right or just one) before showing the shape-analysis images, so as to give a clear idea of the real microstructure analyzed.

lines 187: Table 2: The table is very confusing and difficult to read. The authors should enter the results in a more orderly manner combing the values in a single row not in separate columns and where it is missing, the authors should specify the units used.

Please for each Area (and parameter) combine the Min -Max values (Mean ± D.S).

Author Response

Dear reviewer,

Thank you very much for taking the time to review this manuscript titled "Morphological differences and contour visualization of statoliths in different geographic populations of purpleback flying squid (Sthenoteuthis oualaniensis)". I greatly appreciate your valuable feedback and suggestions. I have carefully considered all of your comments and made revisions to the manuscript accordingly.

Best regards,

Moxian Chu

Reviewer 2 Report

Comments and Suggestions for Authors

Dear authors

The manuscript includes new data and interesting results useful for fisheries and biological issues but needs improvement in content and format.

Please used italics for genus and species names.

In introduction, please change fish statoliths for fish otoliths.

In Methods and Results authors did not include sufficient information on Table and Figure legends. Please improve it including species name and location. Specifically, explain in Fig. 3 which figs correspond to each method (traditional, Fourier, geometric). Include in fig and legend some identifier (e.g., a,b,c) to understand which is which. Tables are difficult to follow, please used small size (8 or 9).

Please include a short paragraph in the discussion about the results of the manuscript before talking about specific issues.

Suggested references:

https://academic.oup.com/mollus/article/88/1/eyab046/6534512

https://scientiamarina.revistas.csic.es/index.php/scientiamarina/article/view/1407

Author Response

(The authors gave the same response as above.)

Reviewer 3 Report

Comments and Suggestions for Authors

Minor improvements suggested. See an attached file

Comments on the Quality of English Language

Simplify your sentences. Finish your words.

Author Response

(The authors gave the same response as above.)
